# Advanced Immunomodulation in Rheumatoid Arthritis: Immune Checkpoints, microRNAs, and Cell-Based Therapies

**DOI:** 10.3390/biomedicines13092186

**Published:** 2025-09-07

**Authors:** Sandra Pascual-García, Raúl Cobo, José Luis Bolinches, Iván Ortiz, Pedro Viamonte, José Miguel Sempere-Ortells, Pascual Martínez-Peinado

**Affiliations:** Department of Biotechnology, University of Alicante, 03690 San Vicente del Raspeig, Spain

**Keywords:** rheumatoid arthritis, immune checkpoints, microRNAs, cell-based therapies, personalised medicine

## Abstract

**Background/Objectives**: Rheumatoid arthritis (RA) is a chronic autoimmune disorder marked by persistent synovial inflammation, progressive joint destruction, and systemic complications. Despite significant progress in targeted therapies, major clinical challenges persist, including heterogeneous treatment responses and therapeutic resistance. This review aims to critically evaluate emerging immunomodulatory strategies—focusing on immune checkpoints, microRNAs (miRNAs), and cell-based therapies—as potential diagnostic and therapeutic tools. **Methods**: This non-systematic literature review involved a comprehensive analysis of recent studies to investigate emerging immunomodulatory strategies in RA. Special attention was given to immune checkpoint pathways—cytotoxic T-lymphocyte antigen 4 (CTLA-4); programmed death-1 (PD-1) and its ligand, PD-L1; and inducible T-cell costimulator (ICOS)—as well as cell-based therapies. Additionally, miRNA-based interventions were examined for their diagnostic and therapeutic potential. **Results**: Immune checkpoint modulation has demonstrated preclinical efficacy in attenuating inflammatory responses and restoring immune tolerance. Concurrently, miRNAs have emerged as both biomarkers and therapeutic agents, with exosome-based delivery systems enhancing their function. Cell-based therapies have shown robust immunoregulatory effects with acceptable safety profiles. Notably, integrative strategies that combine checkpoint inhibitors, cell-based interventions, and miRNA delivery exhibit synergistic effects and offer a promising avenue for personalised treatment, when guided by molecular and transcriptomic profiling. The majority of these approaches remain at the preclinical or early translational stage. **Conclusions**: Targeted immunomodulation is poised to transform RA management. The integration of cell therapies, checkpoint inhibition, and miRNA manipulation with omics technologies holds promise for enhancing therapeutic precision and safety. Advancing towards personalised immunotherapy will necessitate a multidisciplinary and patient-centred effort.

## 1. Introduction

Rheumatoid arthritis (RA) is a chronic, systemic autoimmune disease that affects approximately 0.5% of the global population [1]. It typically manifests around the age of 45 and occurs three times more frequently in women than in men [2,3]. RA is characterised by persistent inflammation of the synovial membrane, which leads to progressive destruction of cartilage and bone, joint deformities, and functional disability [4]. In addition to articular damage, RA may also result in extra-articular manifestations, including cardiovascular disease, osteoporosis, and interstitial lung involvement [5,6].

Clinically, RA presents with joint pain, prolonged morning stiffness, swelling, and functional impairment, primarily affecting the small joints of the hands and feet [7]. Over time, chronic inflammation results in irreversible structural damage, significantly compromising patients’ quality of life.

Diagnosis is based on the 2010 classification criteria established by the American College of Rheumatology (ACR) and the European League Against Rheumatism (EULAR), which consider the number and type of affected joints, serological markers—such as rheumatoid factor (RF) and anti-cyclic citrullinated peptide antibodies (ACPA)—and acute-phase reactants, including C-reactive protein (CRP) and erythrocyte sedimentation rate (ESR) [8]. Imaging techniques such as musculoskeletal ultrasound and magnetic resonance imaging (MRI) enhance diagnostic sensitivity by enabling the detection of subclinical synovitis and early bone erosions [9,10].

In terms of treatment, disease-modifying antirheumatic drugs (DMARDs) remain the cornerstone of RA management, with methotrexate as the first-line conventional synthetic DMARD (csDMARD) and biologics (bDMARDs) or Janus kinase (JAK) inhibitors, also known as targeted synthetic DMARDs (tsDMARDs), reserved for patients with inadequate response [11]. Current guidelines emphasise methotrexate as the anchor drug and support early treatment individualisation through shared decision-making. Recent safety concerns regarding JAK inhibitors, particularly after a safety study in a clinical trial of tofacitinib versus tumour necrosis factor (TNF) inhibitor in subjects with rheumatoid arthritis, also referred to as the ORAL Surveillance trial, have led regulatory agencies to issue restrictions and boxed warnings, highlighting the need for careful patient selection [12].

The aetiology of RA is multifactorial, involving a complex interplay between genetic predisposition, epigenetic modifications, and environmental factors such as smoking and alterations in the microbiome [13,14,15,16,17]. This pathogenic complexity contributes to the marked clinical heterogeneity of RA, which poses challenges for early diagnosis and personalised treatment strategies [18]. The immune system plays a central role in RA pathogenesis [17], and in recent decades, significant progress has been made in elucidating the underlying immunological mechanisms and in the development of targeted immunomodulatory therapies.

### Immune Cell Activation and Loss of Tolerance in RA

RA originates from immunological dysregulation that results in a loss of tolerance to self-antigens and chronic activation of immune cells [19]. Within inflamed synovial tissue, CD4^+^ T lymphocytes—particularly the T helper 1 (Th1) and Th17 subsets—play a central role by activating macrophages, inducing B cell proliferation, and promoting the release of pro-inflammatory cytokines such as TNF-α, interleukin-1β (IL-1β), and IL-6 [20,21].

Regulatory T (Treg) cells, which under physiological conditions regulate autoimmunity and maintain immune homeostasis, display impaired functionality in patients with RA [17]. This dysfunction allows sustained activation of effector cells and perpetuates synovial inflammation and tissue damage [22].

In parallel, a substantial body of evidence supports the critical role of fibroblast-like synoviocytes (FLSs) in the pathogenesis of inflammatory joint disease. These cells not only respond to inflammatory stimuli but also actively contribute to the perpetuation of inflammation and joint destruction [23,24]. FLSs produce matrix metalloproteinases (MMPs) and other degradative enzymes that degrade extracellular matrix components, activate osteoclasts, and invade adjacent cartilage, ultimately leading to its destruction [25,26].

In addition, microRNAs (miRNAs) have emerged as important regulators of immune responses in RA. These small non-coding RNAs modulate gene expression at the post-transcriptional level and are implicated in key processes such as Th17 cell differentiation, the imbalance between osteoclast and osteoblast activity, and the regulation of FLS function [27,28,29]. 

This review provides a critical and up-to-date overview of emerging therapeutic strategies in RA, with a particular emphasis on immune checkpoints, cell-based therapies, and miRNAs. The roles of these novel approaches in modulating immune responses and their translational potential for clinical application are discussed in detail.

## 2. Materials and Methods

This article is a traditional narrative review and does not follow a systematic review protocol. The literature search was performed in PubMed, Scopus, and Google Scholar, focusing primarily on articles published within the last five years. Additional relevant publications outside this time frame were included when considered seminal or highly influential. Both preclinical and clinical studies were considered, with preference given to peer-reviewed original research and review articles in English.

During the preparation of this manuscript/study, the authors used ChatGPT (GPT-4-turbo) for the purposes of refining the language and clarity of the text and for generating the image of the inflamed joint included in Figure 1 and Figure 2.

## 3. Immune Checkpoint Modulation in RA

Immune checkpoints are crucial regulatory molecules that modulate T cell activation and peripheral tolerance, thereby preventing autoimmune responses and preserving immune homeostasis [30]. Their significance lies in their dual function: while they suppress excessive immune activity, they may also be exploited by tumours and chronic inflammatory diseases to evade immune surveillance [30]. Among the most studied immune checkpoints are cytotoxic T-lymphocyte antigen 4 (CTLA-4); programmed death-1 (PD-1) and its ligand, PD-L1; and inducible T-cell costimulator (ICOS), all of which are under active investigation in the context of autoimmune disorders, including RA (Figure 1).

### 3.1. CTLA-4

CTLA-4 is an inhibitory receptor expressed on activated T lymphocytes and Treg cells which competes with the costimulatory molecule CD28 for binding to CD80/CD86 ligands on antigen-presenting cells [31]. In contrast to CD28, which enhances T cell activation, CTLA-4 transmits inhibitory signals that limit T cell proliferation and cytokine secretion [32].

In RA, the expression and function of CTLA-4 are reduced, contributing to impaired Treg-mediated suppression and aberrant immune activation [33]. Among checkpoint-targeted strategies, CTLA-4 modulation is the only approach currently established in RA clinical care. Abatacept, a CTLA-4–Immunoglobulin (Ig) fusion protein, acts by blocking the CD28–CD80/CD86 interaction, thereby modulating T cell costimulation [34]. It has been demonstrated that abatacept reduces synovial inflammation, inhibits osteoclast differentiation, prevents joint destruction, and modulates Th17/Treg balance through regulation of forkhead box P3 (FoxP3) and retinoic acid receptor-related orphan receptor gamma t (RORγt) transcription factors [35,36]. Furthermore, abatacept therapy is associated with a decrease in Th17 and Th1 cell frequencies in RA patients [37].

### 3.2. PD-1/PD-L1

The PD-1/PD-L1 axis plays a pivotal role in the induction and maintenance of peripheral tolerance. Engagement of PD-1 by its ligand, PD-L1, inhibits T cell proliferation and enhances the production of anti-inflammatory cytokines, thus preventing excessive immune activation [38,39].

In RA, elevated PD-1 expression on T lymphocytes has been observed, particularly in patients with active disease [40]. Moreover, increased levels of soluble PD-L1 correlate with inflammatory disease activity [41]. PD-1^+^ CD8^+^ T cells have also been implicated in promoting the differentiation of memory B cells into plasmablasts via IL-21 signalling [42]. 

However, considerable controversy remains regarding the role of these molecules in RA. For instance, experimental models have demonstrated that enhancement of PD-L1 reduces joint inflammation, highlighting the potential of this pathway as a therapeutic target and disease biomarker [43]. In humans, evidence is still confined to early-phase studies. A phase 2 clinical trial of the agonistic anti-PD-1 monoclonal antibody peresolimab reported improvements in disease activity score (DAS) 28-CRP, suggesting that stimulation of the PD-1 receptor may attenuate disease activity in RA [44]. While these results are encouraging, confirmation in larger, late-phase trials is required. 

Beyond peresolimab, several other PD-1 agonists are under active development for RA. Rosnilimab, a PD-1 agonist IgG1 monoclonal antibody, has been shown to help re-establish immune balance by attenuating autoreactive T cell activity in a phase 2b clinical trial [45]. JNJ-67484703, a humanised IgG1κ PD-1 agonist/depleter, was evaluated in a randomised, double-blind, placebo-controlled phase 1b study in patients with active RA [46]. After 10 weeks of subcutaneous dosing (2 mg/kg and 3 mg/kg), it was generally safe and well-tolerated, and the 3 mg/kg group demonstrated numerically greater improvements in DAS28-CRP at week 12, along with selective depletion of circulating PD-1^+^ T cells and low immunogenicity [46]. Collectively, these data highlight the therapeutic promise of PD-1 agonism in RA but also emphasise that long-term safety, durability, and comparative efficacy remain to be defined.

Interestingly, this stands in sharp contrast to the oncology setting, where systemic PD-1/PD-L1 blockade with immune checkpoint inhibitors (e.g., nivolumab and pembrolizumab) enhances anti-tumour immunity but frequently induces immune-related adverse events (irAEs), such as inflammatory arthritis [47]. These effects arise because inhibition of PD-1 signalling removes a critical checkpoint of peripheral tolerance, resulting in excessive T cell activation and loss of self-tolerance [47,48]. By contrast, in RA, therapeutic activation of PD-1/PD-L1 signalling may restore these inhibitory pathways, thereby dampening autoreactive T cell responses and reducing synovial inflammation. This apparent paradox underscores the context-dependent role of the PD-1/PD-L1 axis, with agonism being beneficial in autoimmunity while inhibition enhances anti-tumour responses.

### 3.3. ICOS

ICOS is expressed on activated T cells, including Treg cells and follicular helper T (Tfh) cells, and is critical for the differentiation of Th17 lymphocytes [49,50]. In contrast to PD-1/PD-L1 and CTLA-4, where agonistic approaches have therapeutic benefit, blockade of ICOS or its ligand (ICOSL) reduces disease severity in murine models of rheumatoid arthritis, such as the glucose-6-phosphate isomerase (GPI)-induced arthritis model [51]. However, the available evidence is currently restricted to preclinical models.

Patients with RA exhibit elevated ICOS expression on T lymphocytes within the synovial fluid, suggesting a role in promoting the inflammatory microenvironment [52]. Additionally, ICOS expression on CD19^+^ B cells is associated with symptom severity both in patients and in the collagen-induced arthritis (CIA) mouse model [53]. Interestingly, increased frequencies of ICOS^+^ Treg cells have also been detected in RA patients and negatively correlate with disease activity, indicating a possible protective function and supporting the utility of ICOS^+^ Treg cells as potential markers of remission [54].

## 4. Immune Regulation and Clinical Applications of miRNAs in RA

miRNAs are single-stranded RNA molecules of approximately 20 nucleotides in length [55], whose primary function is the post-transcriptional regulation of gene expression [56]. Their biogenesis begins in the nucleus, where RNA polymerase II transcribes specific genomic regions to generate primary transcripts, known as primary (pri-)miRNAs [57]. The DiGeorge syndrome critical region 8 (DGCR8) protein facilitates the recruitment of the RNase III enzyme Drosha, which cleaves the pri-miRNA to produce a precursor miRNA (pre-miRNA) of approximately 70 nucleotides [58,59,60,61,62].

These pre-miRNAs are then exported to the cytoplasm by Exportin-5 [61]. In the cytoplasm, Dicer, in complex with the transactivation response RNA-binding protein (TRBP), further processes the pre-miRNA into a double-stranded intermediate [62,63]. One strand— referred to as the passenger strand, also denoted as miRNA*—is subsequently degraded, while the guide strand is incorporated into the RNA-induced silencing complex (RISC) [64]. The RISC complex, which includes Argonaute (AGO) proteins [65,66], mediates gene silencing by binding to complementary sequences on target mRNAs [67]. In this section, we examine miRNAs with potential as biomarkers and immunoregulatory agents in RA (Figure 2).

### 4.1. Pro-Inflammatory and Anti-Inflammatory miRNAs

Recent studies in patients with RA have identified elevated levels of several specific miRNAs—including let-7d, miR-24, miR-126, miR-130a, miR-221, and miR-431—which may serve as potential diagnostic or prognostic biomarkers [68]. At present, such findings remain exploratory and have not yet been translated into validated clinical tools. 

Functional characterisation of these miRNAs has revealed diverse roles, with certain miRNAs exhibiting pro-inflammatory properties (e.g., miR-155), while others, such as miR-146a, display predominantly immunoregulatory effects [69,70]. Notably, some miRNAs, including miR-21, possess both pro- and anti-inflammatory activities depending on cellular context [71,72,73].

Among the pro-inflammatory miRNAs, miR-21, miR-26, and miR-155 are particularly prominent [74]. These miRNAs contribute to the activation of Th17 cells, memory Treg cells, and B lymphocytes, as well as to the production of IL-17 and the pathological activity of FLSs [74]. Specifically, miR-21 expression is reduced in patients with juvenile idiopathic arthritis (JIA); however, it has been shown to inhibit nuclear factor kappa B (NF-κB) signalling and promote macrophage polarisation towards the anti-inflammatory M2c phenotype [75,76]. Similarly, miR-155 promotes the production of IL-1β, IL-6, and TNF-α by FLSs and contributes to joint inflammation and destruction [69,77].

Members of the miR-26 family also participate in RA pathogenesis. For instance, miR-26b suppresses FLS proliferation, invasion, and migration, while downregulating TNF-α, IL-1β, and IL-6 expression [78,79]. In contrast, miR-26a exhibits a notable anti-inflammatory role by inhibiting osteoclast differentiation and function, thereby limiting osteoclastogenesis [75,76]. 

Conversely, miR-146a is among the most extensively characterised anti-inflammatory miRNAs [70]. It modulates the secretion of TNF-α, IL-6, and IL-17 by Th17 cells [27]. Although elevated levels of miR-146a are associated with increased RA disease activity, they inversely correlate with Treg frequency [80], suggesting a compensatory regulatory mechanism. Another miRNA with similar properties is miR-124, which is found at reduced levels in the synovial fluid, plasma, and peripheral blood mononuclear cells (PBMCs) of RA patients [81]. Experimental modulation of miR-124 expression enhances anti-inflammatory cytokine production, increases Treg abundance, and reduces Th17 cell frequency and pro-inflammatory cytokine levels [81]. These observations are promising but remain preclinical and have not yet advanced into clinical trials.

### 4.2. Diagnostic, Prognostic, and Therapeutic Response Biomarkers

Circulating miRNAs are remarkably stable in biological fluids and have demonstrated considerable diagnostic and prognostic utility in autoimmune diseases. In RA, elevated levels of miR-16, miR-146a, miR-155, and miR-223 have been detected in synovial fluid when compared to patients with non-inflammatory joint disorders [82,83]. Although consistently observed in patient cohorts, these signatures remain investigational and are not currently implemented in clinical practice.

Among these, miR-223 plays a particularly important role in osteoclastogenesis and bone erosion [84]. It also promotes T lymphocyte infiltration into synovial tissue, contributing to the chronic inflammatory state [84]. In contrast, miR-210 has been reported to negatively correlate with pro-inflammatory cytokine levels and with the severity of clinical symptoms in RA [77], highlighting its potential as a biomarker of disease activity and therapeutic response. Further validation in large prospective cohorts will be required before these molecules can be considered clinically actionable. 

Taken together, current evidence suggests that only a subset of miRNAs—particularly miR-146a, miR-155, and miR-223—have been consistently validated across RA patients, making them the most promising for clinical translation. In contrast, miRNAs such as miR-124 and miR-210 show anti-inflammatory potential but remain largely confined to exploratory or small-scale studies, while others (e.g., the miR-26 family) are restricted to preclinical investigations. This stratification highlights that although numerous miRNAs are mechanistically relevant, only a few currently stand as reproducible biomarker candidates, and none have yet reached clinical application.

## 5. Immunomodulatory Cell-Based Therapies

Immunomodulatory cell-based therapies represent a promising frontier in the treatment of RA, aiming to restore immune tolerance and attenuate chronic inflammation [85]. This section discusses emerging strategies, including chimeric antigen receptor (CAR) T cells, induced Treg cells, tolerogenic dendritic cells (tolDCs), and mesenchymal stromal cells (MSCs)—each offering novel approaches to rebalancing immune responses in RA (Table 1).

### 5.1. CAR-T Cells

CAR-T cells, initially developed for oncology, are now being investigated in autoimmune diseases such as RA [86]. These genetically engineered T cells are designed to selectively eliminate autoreactive lymphocytes that drive the pathogenic immune response [86]. In murine models, in vitro CD8^+^ CAR-T cells directed against specific CD4^+^ T cells have successfully eliminated pathogenic Th cells, reduced inflammation, and limited autoantibody production [87]. This strategy provides targeted immunosuppression without compromising overall immune competence, which would represent a significant therapeutic advantage. Recently, a first-in-human study reported the use of autologous, fourth-generation CD19-targeted CAR-T cells engineered to secrete antibodies against IL-6 and TNF-α in patients with difficult-to-treat RA [88]. This approach could not only deplete pathogenic B cells but also mitigate cytokine-driven toxicities, directly addressing one of the major safety concerns. Nonetheless, until larger clinical studies confirm both its efficacy and safety, CAR-T therapy in RA should be regarded as highly experimental.

### 5.2. Treg Cells

Treg cells play a fundamental role in maintaining immune homeostasis; however, both their frequency and suppressive function are diminished in RA [98]. In CIA models, the adoptive transfer of induced Treg cells has been shown to reduce inflammation and protect against joint damage [89]. Notably, collagen type II-specific Tr1 cells (Col-Treg cells) mediate immunosuppression via secretion of anti-inflammatory cytokines and have been demonstrated to alleviate disease symptoms in CIA mice [90]. 

Moreover, transfer of induced human Treg cells into CIA models results in decreased disease severity, reduced B cell activation, and lower levels of pro-inflammatory cytokines [91]. In vitro, this intervention also leads to reduced proliferation and increased apoptosis of rheumatoid arthritis synovial fibroblasts (RASFs) [91], further supporting the therapeutic potential of Treg-based strategies. Despite these promising results, evidence remains confined to preclinical models, and clinical translation in RA patients has yet to be achieved.

### 5.3. TolDCs

TolDCs are monocyte-derived cells differentiated in the presence of immunomodulatory factors such as autologous synovial fluid [93]. Their principal role is to induce immune tolerance by promoting Treg development and suppressing effector T cell responses [92]. Preclinical studies have shown that tolDCs can modulate both naïve and effector CD4^+^ T cells, enhance Treg proliferation, and attenuate synovial inflammation and joint destruction in murine arthritis models [92]. 

Early clinical data are beginning to emerge. A phase I trial assessed intra-articular infusion of tolDCs in RA patients. While overall feasibility and safety were confirmed, stabilisation of knee symptoms was only observed in 2 of the 13 recruited patients—specifically those receiving the highest doses (10^7^ tolDCs). These findings underscore the need for larger, controlled studies before clinical application can be considered [93].

A further limitation is the variability in tolDC generation protocols, which results in heterogeneous immunoregulatory capacity across studies [99,100]. This lack of standardisation complicates reproducibility and would represent a significant barrier to clinical development.

### 5.4. MSCs and Extracellular Vesicles

MSCs are multipotent, self-renewing cells capable of differentiating into osteoblasts, chondrocytes, and adipocytes [101]. Beyond their regenerative capabilities, MSCs possess potent immunomodulatory functions [101], making them promising candidates for the treatment of autoimmune diseases, including RA.

In a preclinical study, a triple therapy combining oligosaccharides, human placental extract, and rat bone marrow-derived MSCs exhibited strong anti-inflammatory and anti-arthritic effects in a rat model of complete Freund’s adjuvant (CFA)-induced arthritis [94]. This regimen significantly reduced serum levels of IL-6, IL-10, and TNF-α, without adversely affecting haematological, hepatic, renal, or endocrine parameters [94]. Histopathological analysis demonstrated enhanced bone healing and reduced apoptosis, indicating improved tissue regeneration [94].

In humans, evidence remains limited. A phase I/IIa clinical trial evaluated a single intravenous infusion of autologous adipose-derived MSCs in 15 patients with active RA [95]. While no significant changes were observed in systemic inflammatory markers, the treatment led to improvements in joint swelling and tenderness, as measured by ACR66/68 scores, suggesting that these MSCs may safely ameliorate articular symptoms in RA [95]. Similarly, in another phase Ib/IIa clinical trial, the infusion of allogeneic adipose-derived MSCs in patients with refractory RA was well tolerated, with few adverse events reported [96]. These early-phase human studies confirm the feasibility and short-term safety of MSC therapy in RA; however, they were not sufficiently powered to assess efficacy, highlighting the need for larger, controlled trials.

Further research has explored the utility of MSC-derived small extracellular vesicles (MSC-sEVs) as a cell-free alternative [97]. In a CIA mouse model, MSC-sEV treatment resulted in decreased disease severity; lower levels of anti-type II collagen antibodies, IL-6, and C5b-9; and promoted a shift in macrophage polarisation towards an anti-inflammatory phenotype (increased M2, decreased M1) [97]. Histological assessments confirmed reductions in synovial inflammation, pannus formation, cartilage destruction, and bone erosion [97]. Although MSC-sEVs were less effective than prednisolone, they represent a promising, less toxic class of emerging bDMARDs for RA treatment [97]. However, these results are restricted to preclinical models, and no clinical trials have yet been reported. 

While preclinical and early-phase trials suggest that MSC therapy would be safe in the short term, unresolved issues include their short persistence, potential senescence, and theoretical risks of malignant transformation [102,103]. Long-term surveillance will therefore be crucial to establish the safety profile of MSC-based interventions.

## 6. Therapeutic Perspectives and Challenges in RA Immunomodulation

### 6.1. Combination Strategies: Cell-Based Therapy, Immune Checkpoints, and miRNAs

Combined therapeutic strategies aim to overcome the limitations of single-agent interventions. For example, MSCs preconditioned with interferon (IFN)-γ have been shown to reduce clinical signs of inflammation in CIA mouse models, although such evidence remains limited to preclinical research [104]. Clinical studies suggest that MSC therapy may improve outcomes in RA patients; however, the therapeutic response appears to vary depending on the dose, timing, and patient-specific factors [96]. The latter study, which is still in early clinical phases, indicates that the highest dose (4×10^6^ Ad-MSCs/kg) produces slightly greater efficacy than placebo, though ACR70 responses remain consistently low across all cohorts [96]. Additionally, three months after MSC infusion, approximately 54% of patients exhibit a good or moderate clinical response [105]. Collectively, these findings support a potential clinical benefit for MSC-based therapies, albeit with variable efficacy.

Genetically engineered, antigen-specific Treg cells offer targeted immunosuppression by reducing pro-inflammatory Th17 cell populations and preventing bone damage, although such approaches have so far only been validated in preclinical models [106]. In parallel, novel immune checkpoint targets—such as lymphocyte-activation gene 3 (LAG-3), T-cell immunoglobulin and mucin domain-containing proteins (TIM-1 and TIM-3), and T-cell immunoreceptor with Ig and ITIM domains (TIGITs)—are under investigation. Early-phase human RA data have shown that increased plasma and synovial levels of soluble LAG-3 in RA patients correlate with joint inflammation, bone erosion, and elevated levels of pro-inflammatory cytokines [107]. In preclinical murine models, blockade of TIM-1 and TIM-3 leads to reduced levels of IL-6 and IL-17A, diminished CD4^+^ T cell infiltration, and improved clinical outcomes [108]. Moreover, TIGIT inhibition has been shown to enhance natural killer (NK) cell function in RA patients, although these data have not yet been translated into interventional trials, thus belonging to the category of early-phase human RA data [109].

MSCs engineered to overexpress miR-146a and deliver sEVs have been found to increase FoxP3 expression in Treg cells and promote elevated levels of transforming growth factor (TGF)-β and IL-10 in CIA mice, once again representing preclinical advances [110]. In another approach, tolDCs—generated by treating monocyte-derived dendritic cells with dexamethasone, vitamin D3, and lipopolysaccharide—have demonstrated potent immunoregulatory effects on CD4^+^ T cells via TGF-β1 signalling [111]. Although CD4^+^ T cells from RA patients often display reduced responsiveness to TGF-β1, they remain effectively modulated by tolDCs, highlighting their therapeutic promise supported by early-phase human RA data [111].

Nanoparticle-based delivery systems also offer innovative therapeutic strategies. Co-delivery of the non-steroidal anti-inflammatory drug ketoprofen and miR-124 via nanoparticles has demonstrated enhanced anti-inflammatory effects and superior efficacy in a preclinical adjuvant-induced arthritis rat model [112]. Similarly, delivery of miR-23b-loaded nanoparticles induces macrophage apoptosis and suppresses pro-inflammatory pathways, also demonstrated only in preclinical work [113]. Further preclinical studies have shown that exosomes derived from M2 macrophages, co-loaded with IL-10 plasmid DNA and betamethasone, preferentially target inflamed joints in murine models, promoting M1-to-M2 macrophage repolarisation and reducing levels of inflammatory cytokines [114].

It is important to note that most of the evidence discussed in this review originates from preclinical models, particularly murine CIA- and GPI-induced arthritis. While these models are invaluable for mechanistic insights, their predictive value for human disease is limited. For example, multiple interventions that showed efficacy in CIA—including adoptive Treg transfer and MSC-derived vesicles—have demonstrated only modest or inconsistent effects in early clinical studies. Moreover, interspecies differences in immune cell subsets, cytokine networks, and pathophysiology contribute to a high attrition rate in translation to clinical trials [115,116]. Regulatory and logistical barriers, particularly for cell-based therapies, would further delay clinical implementation. Realistically, the widespread adoption of strategies such as CAR-T cells, engineered MSCs, or miRNA delivery systems is likely to require several years of additional research and safety validation before clinical integration can be achieved. 

### 6.2. Current Limitations of Traditional Therapies

Although targeted therapies have transformed RA management, significant challenges persist. Approximately 20% of patients experience inadequate response or treatment failure, a group now recognised as being affected by "difficult-to-treat" RA [117]. Furthermore, while DMARDs are effective, their use is limited by safety concerns, such as increased infection risk, hypersensitivity reactions, and malignancy [118].

A major cause of reduced efficacy is immunogenicity, particularly the development of anti-drug antibodies (ADAbs), which occurs in up to one-third of patients receiving biologics [119]. ADAbs are especially prevalent with anti-TNF-α monoclonal antibodies and are influenced by treatment regimens and individual immune profiles. This problem, well-documented in clinical cohorts, represents evidence established in RA care, although detection methods for ADAbs remain inconsistent, limiting their clinical utility [119]. In addition, the plasticity of inflammatory pathways often allows alternative signalling cascades to compensate when one is therapeutically inhibited [120].

Pharmacogenomic studies have identified genetic polymorphisms associated with differential treatment responses. Variants in Fc fragment of IgG receptor IIIa (*FCGR3A*), *NF-κB*, and Toll-like receptor 4 (*TLR4*) genes correlate with responses to bDMARDs, while *IL-6R* variants are linked to adverse events in patients treated with tocilizumab [121,122,123]. These insights derive from early-phase human RA data that are not yet standardised for routine practice. Similarly, in an early-phase human RA study, elevated levels of miR-5196 have been observed in RA patients prior to anti-TNF-α therapy and are reduced following treatment, suggesting a role as a predictive biomarker [124]. However, these findings have yet to be integrated into routine clinical practice.

A further limitation of current therapies is their limited regenerative capacity. Radiographic progression may continue even during apparent clinical remission [125]. Interestingly, maintaining therapy following progression appears to confer greater benefits than intensification, as fewer patients lose disease control over time [125].

### 6.3. Towards Personalised Immunotherapy in RA

Personalised medicine in RA is being advanced through the integration of clinical and molecular biomarkers, with single-cell transcriptomic technologies playing a pivotal role. Single-cell RNA sequencing (scRNA-seq) of PBMCs from RA patients has identified 18 immune cell subsets, revealing disease-associated signatures such as IFN-induced transmembrane 3 (IFITM3)-expressing monocytes, increased CD4^+^ effector memory T cells in active disease, and reduced nonclassical monocytes during remission [126]. These observations are derived from early-phase human RA data. In synovial fluid, scRNA-seq has detected pathogenic cell populations—macrophages expressing secreted phosphoprotein 1 (SPP1) and CD4^+^ T cells expressing CXC motif chemokine ligand 13 (CXCL13)—that diminish after treatment with anti-TNF-α agents or JAK inhibitors [127]. These findings are accompanied by therapy-specific changes in cellular signalling and interactions [127], highlighting the potential of scRNA-seq to predict therapeutic response.

In FLSs from a TNF-α-induced preclinical mouse model of RA, single-cell analysis revealed that even after histological resolution, FLSs retained inflammatory signatures and secreted endothelial-disrupting factors, suggesting persistent disease activity despite apparent remission [128]. Single-cell and spatial transcriptomics of synovial tissue further identified distinct dendritic cell subsets: tolerogenic AXL receptor tyrosine kinase (AXL)^+^ DC2s in healthy joints and pro-inflammatory DC3s in active RA [129]. Interestingly, tolerogenic DC2s did not recover during remission [129], pointing to a potential therapeutic target for restoring immune tolerance.

Another innovative approach involves the use of digital twins—virtual representations of biological systems that model individual disease dynamics and therapeutic responses [130]. In RA, digital twins may facilitate personalised treatment strategies and optimise therapeutic decision-making. Immune digital twins also hold potential for accelerating drug discovery, although substantial technical and interdisciplinary hurdles must be overcome before widespread clinical adoption [131]. Some studies demonstrated the utility of biomimetic digital twins in elucidating RA pathogenesis by integrating multi-omics data—including exome sequencing and genotype–phenotype associations—to identify both known and novel pathogenic variants [132]. Another study developed a modular biochemical reaction map encompassing over 1000 biomolecules, enabling preclinical in silico simulations of current treatments and the identification of new therapeutic targets [133].

Finally, liquid biopsies—including circulating cell-free DNA, immune cells, and extracellular vesicles—are emerging as valuable tools for non-invasive monitoring and personalised medicine [134]. Already widely applied in oncology, these approaches are now being investigated for their potential to predict treatment response and disease progression in early-phase human RA studies [134].

## 7. Conclusions

Our understanding of RA has progressively evolved towards a multidimensional paradigm, in which immunomodulatory strategies, cell-based therapies, and cutting-edge omics technologies converge to redefine disease management. Advances in elucidating the immunological mechanisms underpinning RA have catalysed the development of innovative approaches—including immune checkpoint modulation, the deployment of immunoregulatory cells, and miRNA manipulation—which may collectively provide novel therapeutic avenues beyond conventional inflammation control.

Despite these advances, several key challenges persist. The clinical heterogeneity of RA, along with primary and secondary resistance to biologic therapies and the limited regenerative capacity of current interventions, continues to compromise patient outcomes. In addition, the emergence of adverse effects associated with newer treatments—such as JAK inhibitors—highlights the need for rigorous safety evaluation in routine clinical practice.

In this context, combinatorial strategies—such as co-administration of tolerogenic cells with immune checkpoint inhibitors, or the targeted delivery of anti-inflammatory miRNAs via exosomes—represent promising avenues to enhance therapeutic efficacy while mitigating adverse effects. Concurrently, the integration of molecular biomarkers, omics platforms, and artificial intelligence is paving the way for personalised medicine, enabling patient stratification, prediction of treatment response, and proactive optimisation of therapeutic interventions.

Nevertheless, the successful implementation of these innovations requires addressing several critical barriers. These include the need for clinical validation across diverse populations, standardisation of therapeutic protocols, equitable access to advanced technologies, and the establishment of robust ethical and regulatory frameworks. Moreover, comprehensive monitoring of comorbidities and continuous pharmacovigilance will be essential to support a holistic and sustainable care model.

In sum, current evidence supports a paradigm shift towards personalised immunomodulation in RA—an approach aimed not only at controlling inflammation but also at preventing structural joint damage and substantially improving patient quality of life. The future of RA management will depend on sustained multidisciplinary collaboration, technological integration, and an unwavering commitment to patient-centred care.

## Figures and Tables

**Figure 1 biomedicines-13-02186-f001:**
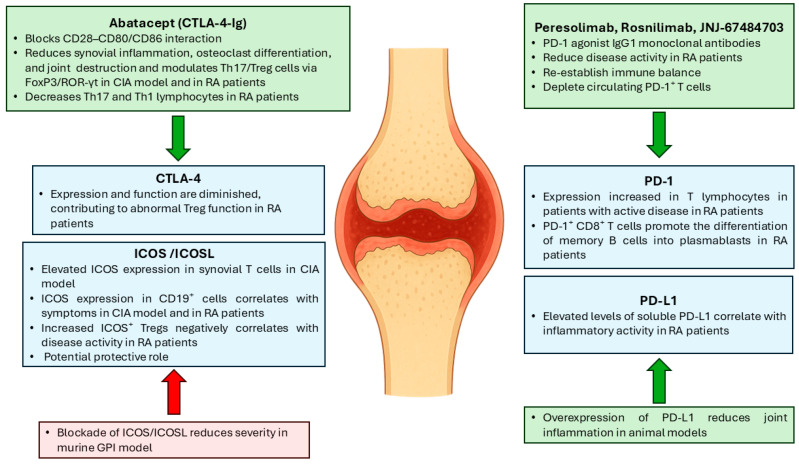
Schematic of the changes in the expression of different immune checkpoints in rheumatoid arthritis (RA). The green arrows and boxes indicate the effects of immune checkpoint agonists in mouse models or RA patients, whereas the red arrows and boxes represent the effects of immune checkpoint blockade. Collagen-induced arthritis (CIA); cytotoxic T-lymphocyte Antigen 4 (CTLA-4); forkhead box P3 (FoxP3); glucose-6-phosphate isomerase (GPI); inducible T-cell costimulator (ICOS); Inducible T-cell costimulatory ligand (ICOSL); immunoglobulin (Ig); interleukin (IL); Johnson & Johnson (JNJ); programmed death-1 (PD-1); PD-1 ligand (PD-L1); retinoic acid receptor-related orphan receptor gamma t (RORγT); T helper cell (Th); regulatory T (Treg) cells. Source: original.

**Figure 2 biomedicines-13-02186-f002:**
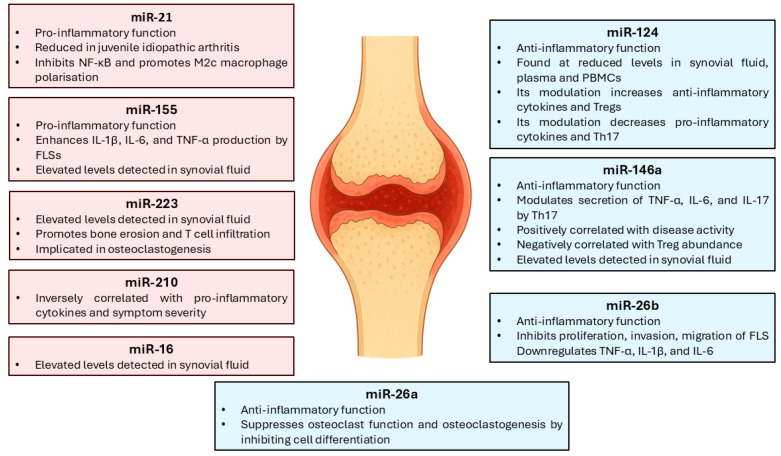
Schematic of the changes in the expression of different microRNAs (miRNAs) in RA. The red boxes indicate the effects of pro-inflammatory miRNAs, whereas the blue boxes illustrate the effects of anti-inflammatory miRNAs in RA. Fibroblast-like synoviocytes (FLSs); nuclear factor kappa B (NF-κB); peripheral blood mononuclear cells (PBMCs); tumour necrosis factor alpha (TNF-α). Source: original.

**Table 1 biomedicines-13-02186-t001:** Immunomodulatory cell-based therapies in RA.

Therapy	Mechanism in RA (Summary)	Experimental Context	References
CAR-T cells	Selectively eliminate autoreactive CD4^+^ T cells; suppress Th responses, inflammation, autoantibody production, and deplete pathogenic B cells without global immunosuppression.	Murine models	[86,87,88]
Treg cells	Attenuate inflammation and joint pathology. Suppress immune activation via anti-inflammatory cytokine release; improve clinical signs.	CIA models and in vitro assays	[89,90,91]
	Reduce disease severity, B cell activation, and cytokine production; inhibit RASF proliferation and induce apoptosis.		
TolDCs	Induce Treg cells and modulate naïve/effector CD4^+^ T cells; reduce joint damage.	Synovial fluid-derived cells in murine arthritis models and phase I clinical trial in RA	[92,93]
MSC-based therapy	Anti-inflammatory and regenerative effects; reduction in IL-6, IL-10, and TNF-α; improved bone repair and tissue survival.	CFA-induced arthritis in rats	[94,95,96]
	Clinically reduce joint swelling and tenderness without altering systemic inflammation; well tolerated.	Phase I/IIa clinical trial in RA	
MSC-derived small extracellular vesicle therapy	Alleviate clinical and histological severity; reduce cytokines and complement activation; promote anti-inflammatory macrophage phenotype.	CIA mouse model	[97]

Different therapies (CAR-T cells, induced Treg cells, tolerogenic dendritic cells, MSC-based therapy, and MSC-derived extracellular vesicle therapy), the mechanisms by which they exert modulatory effects in RA, and the experimental models in which they have been tested are shown. Chimeric antigen receptor T cells (CAR-T cells); complete Freund’s adjuvant (CFA); mesenchymal stromal cells (MSCs); rheumatoid arthritis synovial fibroblasts (RASFs); tolerogenic dendritic cells (TolDCs).

## Data Availability

No new data were created or analysed in this study.

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
