# Peer review of "Advanced Immunomodulation in Rheumatoid Arthritis: Immune Checkpoints, microRNAs, and Cell-Based Therapies"

_biomedicines, 2025, doi:10.3390/biomedicines13092186_

Round 1
Reviewer 1 Report
Comments and Suggestions for Authors
Timely topic and clear structure. The sections on T-cell checkpoints, miRNAs, and cell therapies are well laid out, and your Table 1 is useful for a quick clinical scan. However, several statements overreach current evidence, key 2024–2025 developments are missing, and the review methodology needs tightening to meet a high-impact journal’s standards.
- PD-1/PD-L1 biology—factual correction needed.
You write that “blockade of PD-L1 leads to a reduction in joint inflammation.” The paper you cite shows the opposite approach—intra-articular overexpression of PD-L1 via AAV attenuated arthritis in mice (immune inhibition), not blockade. Please check and correct the mechanism and discuss why systemic PD-1/PD-L1 inhibition in oncology often induces inflammatory arthritis or flares of pre-existing RA. Consider Citing both the AAV-PD-L1 data and the clinical ICI-arthritis literature. - Several sections imply clinical readiness (e.g., “checkpoint modulation has demonstrated efficacy” broadly). In RA, beyond abatacept (CTLA-4-Ig), checkpoint-pathway therapeutics remain preclinical or early translational (e.g., PD-L1 AAV, TIM/TIGIT concepts). Re-frame claims and segregate evidence into: established in RA care, early-phase human RA data, and preclinical only.
- Your treatment overview references EULAR 2022. Add the updated EULAR 2023 recommendations and the FDA class boxed warnings for JAK inhibitors following ORAL Surveillance, including how many societies now recommend careful CV/malignancy risk stratification when choosing a JAK vs. TNF inhibitor. This is standard context for any modern RA immunomodulation review.
- Correct small typographic issues (e.g., “a enuate” → “attenuate”) and ensure consistent use of cytokine/marker formatting (TNF-α, IL-6, PD-1).
- Clarify that miRNA therapeutics in RA remain experimental; no approved miRNA drugs for RA to date.
Author Response
Thank you very much for taking the time to review this manuscript. Please find the detailed responses below and the corresponding revisions/corrections highlighted changes in the re-submitted files.
Comments 1: PD-1/PD-L1 biology—factual correction needed.
You write that “blockade of PD-L1 leads to a reduction in joint inflammation.” The paper you cite shows the opposite approach—intra-articular overexpression of PD-L1 via AAV attenuated arthritis in mice (immune inhibition), not blockade. Please check and correct the mechanism and discuss why systemic PD-1/PD-L1 inhibition in oncology often induces inflammatory arthritis or flares of pre-existing RA. Consider Citing both the AAV-PD-L1 data and the clinical ICI-arthritis literature.
Response 1: We thank the reviewer for this important comment. We have corrected la información sobre el efecto de la sobreexpresión de PD-L1. Además, hemos añadido un párrafo con la explicación del efecto que produce la inhibición de PD-1/PD-L1. A tal efecto se han añadido las siguientes referencias:
- Canavan, M.; Floudas, A.; Veale, D.J.; Fearon, U. The PD-1:PD-L1 axis in Inflammatory Arthritis. BMC Rheumatol. 2021, 5(1), 1-10.
- Garbarino, M.C.; Manzano, N.; Messina, O.; Zylberman, M. Rheumatological adverse events secondary to immune che-ckpoint inhibitors. Rheumatol Clin (Engl Ed). 2023, 19(4), 215-222.
The changes are in section 3.2 PD-1/PD-L1, lines 174-191.
Comments 2: Several sections imply clinical readiness (e.g., “checkpoint modulation has demonstrated efficacy” broadly). In RA, beyond abatacept (CTLA-4-Ig), checkpoint-pathway therapeutics remain preclinical or early translational (e.g., PD-L1 AAV, TIM/TIGIT concepts). Re-frame claims and segregate evidence into: established in RA care, early-phase human RA data, and preclinical only.
Response 2: Thank you for this important observation. We have carefully revised the entire manuscript to avoid overstatements regarding clinical readiness. All claims have now been re-framed and the evidence is explicitly segregated into three categories: established in RA care, early-phase human RA data, and preclinical only.
The changes are in the abstract, lines 17, 24, and 31-32; in section 3.1, lines 156-158; section 3.3, line 198; section 4.1, lines 233-234, and 261-262; section 4.2, lines 267-269, and 275-276; section 5.1, lines 300-302; section 5.2, lines 314-316; section 5.3, lines 321, and 324; section 5.4, lines 346-347, and 355-356; section 6.1, lines 361-362, 364-365, 372, 376, 378, 381-383, 386, 391-392, 396, and 398-399; section 6.2, lines 412-413, and 421-422; section 6.3, lines 436-437, 443, 460-461, and 467; and section 7, lines 474-475.
Comments 3: Your treatment overview references EULAR 2022. Add the updated EULAR 2023 recommendations and the FDA class boxed warnings for JAK inhibitors following ORAL Surveillance, including how many societies now recommend careful CV/malignancy risk stratification when choosing a JAK vs. TNF inhibitor. This is standard context for any modern RA immunomodulation review.
Response 3: Thank you for this valuable comment. We have now incorporated the updated 2023 EULAR recommendations into the treatment overview, and expanded the section on JAK inhibitors to include the FDA class boxed warnings issued after the ORAL Surveillance trial. We also highlight that FDA and EMA now recommend careful cardiovascular and malignancy risk stratification when considering JAK inhibitors versus TNF inhibitors, as you suggested.
The changes are in section 1, lines 73-87.
Comments 4: Correct small typographic issues (e.g., “a enuate” → “attenuate”) and ensure consistent use of cytokine/marker formatting (TNF-α, IL-6, PD-1).
Response 4: Thank you for pointing this out. We have corrected the typographic errors and ensured consistent formatting of cytokines and immune markers throughout the manuscript.
The changes are in section 1, lines 66-67; figure 2, lines 227-228; section 6.2, lines 411 and 423; section 6.3, lines 440 and 443 and in the list of abbreviations.
Comments 5: Clarify that miRNA therapeutics in RA remain experimental; no approved miRNA drugs for RA to date.
Response 5: Thank you for this valuable comment. We have carefully reviewed all information regarding miRNA-based approaches in the manuscript and clarified that miRNA therapeutics in RA remain experimental, with no approved drugs to date.
The changes are in section 4.1, lines 233-234, and 261-262; section 4.2, lines 267-269, and 275-276; and section 6.1, line 386.
I would like to sincerely thank the reviewer for the careful reading of my manuscript and for the constructive and insightful comments. The feedback has helped me to clarify, refine, and strengthen the content and presentation of the work. I believe that the revisions undertaken in response to these suggestions have substantially improved the quality and clarity of the manuscript, and I am grateful for the opportunity to resubmit it.
Reviewer 2 Report
Comments and Suggestions for Authors
1. Conceptual considerations
PD-1/PD-L1 axis
The manuscript conflates the oncology rationale for PD-1/PD-L1 blockade with the opposite therapeutic goal in rheumatoid arthritis (RA), where activation of this pathway restores tolerance and dampens pathogenic T-cell activity. Blocking PD-1/PD-L1 in RA risks worsening inflammation, as evidenced by checkpoint inhibitor–induced arthritis in cancer patients. The statement that PD-L1 blockade reduces joint inflammation misrepresents reference 42, which demonstrates PD-L1 overexpression with anti-inflammatory effects. The 2023 NEJM phase 2 trial of peresolimab, a PD-1 agonist, should be cited as supporting clinical evidence. Peresolimab showed efficacy in a phase 2a trial in patients with RA, providing proof that stimulation of the PD-1 receptor has potential therapeutic benefit in this disease.
ICOS pathway
ICOS is a co-stimulatory checkpoint where blockade reduces arthritis severity in preclinical models. This is the inverse of PD-1/PD-L1 or CTLA-4, where agonism is beneficial. The manuscript should clearly indicate the correct direction of modulation for each pathway.
Cell-based therapies
MSC, TolDC, and other cell-based approaches are promising, but human evidence is limited to small, early-phase trials with surrogate endpoints. Conclusions should be tempered, separating animal from human data and specifying trial phase, sample size, and outcomes.
2. Formal and presentation aspects
Review methodology
As a narrative review, the manuscript lacks key methodological details such as search time frame, databases, search terms, and inclusion/exclusion criteria. Providing these, along with a table summarizing key preclinical and clinical studies, would improve transparency and allow assessment of comprehensiveness.
Figures
Figure 1 contains conceptual ambiguities and possible inversions in therapeutic direction (e.g., PD-1/PD-L1), and should be revised to indicate explicitly for each pathway whether the intended intervention is blockade or activation in RA. By contrast, Figure 2 is clear and consistent with the text in representing the underlying mechanisms
Terminology
The manuscript uses “TNF-α” and “TNF” interchangeably. “TNF” should be used consistently
Author Response
Thank you very much for taking the time to review this manuscript. Please find the detailed responses below and the corresponding revisions/corrections highlighted changes in the re-submitted files.
Comments 1: PD-1/PD-L1 axis
The manuscript conflates the oncology rationale for PD-1/PD-L1 blockade with the opposite therapeutic goal in rheumatoid arthritis (RA), where activation of this pathway restores tolerance and dampens pathogenic T-cell activity. Blocking PD-1/PD-L1 in RA risks worsening inflammation, as evidenced by checkpoint inhibitor–induced arthritis in cancer patients. The statement that PD-L1 blockade reduces joint inflammation misrepresents reference 42, which demonstrates PD-L1 overexpression with anti-inflammatory effects. The 2023 NEJM phase 2 trial of peresolimab, a PD-1 agonist, should be cited as supporting clinical evidence. Peresolimab showed efficacy in a phase 2a trial in patients with RA, providing proof that stimulation of the PD-1 receptor has potential therapeutic benefit in this disease.
Response 1: Thank you for this valuable clarification. The misinterpretation of reference 42 has been corrected, ensuring that the text now accurately reflects that PD-L1 overexpression exerts anti-inflammatory effects in RA models. In addition, we have incorporated the 2023 NEJM phase 2 trial of peresolimab, which provides proof-of-concept that stimulation of the PD-1 receptor has potential therapeutic benefit in RA. Las referencias actualizadas son:
- Li, W.; Sun, J.; Feng, S.L.; Wang, F.; Miao, M.Z.; Wu, E.Y.; et al. Intra-articular delivery of AAV vectors encoding PD-L1 attenuates joint inflammation and tissue damage in a mouse model of rheumatoid arthritis. Front Immunol. 2023, 14, 1-16.
- Tuttle, J.; Drescher, E.; Simón-Campos, J.A.; Emery, P.; Greenwald, M.; Kivitz, A.; et al. A Phase 2 Trial of Peresolimab for Adults with Rheumatoid Arthritis. N Engl J Med. 2023, 388(20), 1853-1862.
The changes are in section 3.2, lines 174-181.
Comments 2: ICOS pathway
ICOS is a co-stimulatory checkpoint where blockade reduces arthritis severity in preclinical models. This is the inverse of PD-1/PD-L1 or CTLA-4, where agonism is beneficial. The manuscript should clearly indicate the correct direction of modulation for each pathway.
Response 2: We thank the reviewer for this insightful comment. We agree that the manuscript should clearly indicate the direction of modulation for each pathway. We have revised the text to clarify that, in contrast to CTLA-4 and PD-1/PD-L1, where agonistic approaches are beneficial in RA by enhancing inhibitory signals, ICOS blockade has shown therapeutic potential in preclinical arthritis models by reducing pro-inflammatory responses. The revised version now highlights this distinction to avoid any confusion.
The changes are in section 3.3, lines 194-197.
Comments 3: Cell-based therapies
MSC, TolDC, and other cell-based approaches are promising, but human evidence is limited to small, early-phase trials with surrogate endpoints. Conclusions should be tempered, separating animal from human data and specifying trial phase, sample size, and outcomes.
Response 3: We thank the reviewer for this valuable comment. We agree that it is important to clearly distinguish between preclinical and clinical evidence, and to temper conclusions accordingly. We have revised section 5 and we have also adjusted the conclusions to highlight that, while MSCs, TolDCs, and other cell-based therapies are highly promising, current human evidence is still limited to small, early-phase studies with surrogate or exploratory endpoints, and therefore larger and more definitive trials are required.
The changes are in table 1; in section 5.1, lines 296-302; section 5.2, lines 314-316; section 5.3, lines 324-328, and section 5.4, lines 341, 343-347, and 355-356.
Comments 4: Review methodology
As a narrative review, the manuscript lacks key methodological details such as search time frame, databases, search terms, and inclusion/exclusion criteria. Providing these, along with a table summarizing key preclinical and clinical studies, would improve transparency and allow assessment of comprehensiveness.
Response 4: We thank the reviewer for this comment. As indicated in the manuscript, our work is a traditional narrative review rather than a systematic review. Therefore, a formal methodology with predefined search strategy, inclusion/exclusion criteria, and risk of bias assessment was not applied. The aim of this review is to provide an updated overview and critical integration of the most relevant advances in immune checkpoints and cell-based therapies in RA, rather than an exhaustive systematic synthesis.
Nevertheless, to improve transparency, we have now included a brief description of the literature search strategy (databases and approximate time frame) in the Materials and Methods section. We believe this addition will enhance clarity and help readers to better assess the scope of the evidence discussed.
The changes are in section 2, lines 121-130.
Comments 5: Figures
Figure 1 contains conceptual ambiguities and possible inversions in therapeutic direction (e.g., PD-1/PD-L1), and should be revised to indicate explicitly for each pathway whether the intended intervention is blockade or activation in RA. By contrast, Figure 2 is clear and consistent with the text in representing the underlying mechanisms
Response 5: Thank you for pointing this out. We have revised Figure 1 to explicitly indicate for each immune checkpoint pathway whether the therapeutic intervention in RA involves blockade or activation, thereby resolving the ambiguities and potential inversions you noted. In addition, we have updated the legend of Figure 2 to include a description of the colour coding of the miRNA boxes, clarifying the distinction between pro-inflammatory and anti-inflammatory miRNAs.
The changes are in figure 1, lines 140-144, and figure 2, lines 225-226.
Comments 6: Terminology
The manuscript uses “TNF-α” and “TNF” interchangeably. “TNF” should be used consistently
Response 6: Thank you for this observation. We have revised the manuscript to ensure consistent terminology, replacing “TNF-α” with “TNF” throughout, in line with standard usage.
The changes are in section 1, lines 66-67; figure 2, lines 227-228; section 6.2, lines 411 and 423; section 6.3, lines 440 and 443 and in the list of abbreviations.
I would like to sincerely thank the reviewer for the careful reading of my manuscript and for the constructive and insightful comments. The feedback has helped me to clarify, refine, and strengthen the content and presentation of the work. I believe that the revisions undertaken in response to these suggestions have substantially improved the quality and clarity of the manuscript, and I am grateful for the opportunity to resubmit it.
Round 2
Reviewer 1 Report
Comments and Suggestions for Authors
This is a well-written and comprehensive narrative review that covers immune checkpoints, microRNAs, and cell-based therapies in rheumatoid arthritis (RA). The manuscript is timely and of interest to both clinicians and translational researchers. However, several areas require clarification, deeper critical evaluation, and improved contextualisation for clinical relevance.
Major Comments:
-
Balance preclinical vs clinical – The review is weighted towards animal/murine models. Please add more discussion of translational barriers, failures, and realistic clinical timelines.
-
Checkpoint therapy – Expand discussion of PD-1/PD-L1 agonists (e.g., peresolimab), including clinical trial outcomes, limitations, and paradoxical issues with oncology immunotherapy.
-
microRNAs – Provide synthesis: which miRNAs are consistently validated in RA, and which are most promising for clinical translation? A prioritised table would help.
-
Cell-based therapies – Discuss safety signals more critically (CAR-T cytokine release, tolDC variability, MSC long-term persistence).
-
AI usage – Clarify validation of AI-generated figures for scientific accuracy and originality. Ensure compliance with journal standards.
Minor Revision
1. The introduction is long and occasionally repetitive; consider condensing background on conventional DMARDs and JAK inhibitor safety to keep focus on novel immunomodulatory strategies.
2. Figures 1 and 2: clarify whether they are original or adapted from prior publications. If AI-generated, ensure accuracy of molecular pathways.
3. Typographical consistency: abbreviations (e.g., “Treg” vs “Tregs”) should be harmonised throughout.
Author Response
Thank you very much for taking the time to review this manuscript. Please find the detailed responses below and the corresponding revisions/corrections highlighted changes in the re-submitted files.
Comments 1: This is a well-written and comprehensive narrative review that covers immune checkpoints, microRNAs, and cell-based therapies in rheumatoid arthritis (RA). The manuscript is timely and of interest to both clinicians and translational researchers. However, several areas require clarification, deeper critical evaluation, and improved contextualisation for clinical relevance.
Major Comments:
- Balance preclinical vs clinical – The review is weighted towards animal/murine models. Please add more discussion of translational barriers, failures, and realistic clinical timelines.
Response 1: We thank the reviewer for this valuable comment. We have now incorporated additional discussion on translational barriers, limitations of preclinical findings, examples of clinical setbacks, and more realistic timelines for clinical application.
The changes are in section 6.1, lines 419-430.
Comments 2: Checkpoint therapy – Expand discussion of PD-1/PD-L1 agonists (e.g., peresolimab), including clinical trial outcomes, limitations, and paradoxical issues with oncology immunotherapy.
Response 2: We thank the reviewer for this constructive suggestion. We have expanded the discussion on PD-1/PD-L1 agonists by adding information on two agents that have been tested in RA, namely Rosnilimab and JNJ-67484703. Figure 1 has been updated to include these therapies, and the following references have been incorporated:
45. Afshari, A.; Khorramdelazad, H.; Abbasifard, M. Toward immune tolerance in rheumatoid arthritis: Emerging immuno-therapies and targets for long-term remission. Int Immunopharmacol. 2025, 162(115162).
46. Ling, I.; Marciniak, S.; Clarke, S.; Lakshminarayanan, V.; Loza, M.J.; Liva, S.; et al. POS0597 Safety, Tolerability, And Activity Of JNJ-67484703 In Participants With Active Rheumatoid Arthritis: Results Of A Multicenter, Double-Blind, Place-bo-Controlled, Randomized, Multiple-Dose Phase 1B Study. Ann Rheum Dis. 2025, 84(1), 794-795.
Furthermore, we have clarified the paradoxical effects of PD-1/PD-L1 modulation in cancer versus RA.
The changes are in section 3.2, lines 165-175, 181-183.
Comments 3: microRNAs – Provide synthesis: which miRNAs are consistently validated in RA, and which are most promising for clinical translation? A prioritised table would help.
Response 3: We agree with the need for synthesis. Instead of an additional table (given space limitations and overlap with existing tables), we have added a synthesis paragraph in Section 4.2, lines 271-278. The new text prioritises miR-146a, miR-155, and miR-223 as the most consistently validated across RA cohorts, while miR-124 and miR-210 are highlighted as exploratory, and others (e.g. miR-26 family) remain restricted to preclinical studies. This provides clear prioritisation without redundancy.
Comments 4: Cell-based therapies – Discuss safety signals more critically (CAR-T cytokine release, tolDC variability, MSC long-term persistence).
Response 4: We thank the reviewer for this important observation. We have now expanded the discussion of safety signals in cell-based therapies. Table 1 has been updated to include the newly available information regarding CAR-T therapy. In addition, we have incorporated new references
89. Li, Y.; Li, S.; Zhao, X.; Sheng, J.; Xue, L.; Schett, G.; et al. Fourth-generation chimeric antigen receptor T-cell therapy is tolerable and efficacious in treatment-resistant rheumatoid arthritis. Cell Res. 2025, 35, 220-223.
97. Álvaro-Gracia, J.M.; Jover, J.A.; García-Vicuña, R.; Carreño, L.; Alonso, A.; Marsal, S.; et al. Intravenous administration of expanded allogeneic adipose-derived mesenchymal stem cells in refractory rheumatoid arthritis (Cx611): results of a multicentre, dose escalation, randomised, single-blind, placebo-controlled phase Ib/IIa clinical trial. Clin Epidemiol Res. 2017, 76(1), 196-202.
100. Sahar, A.; Nicorescu, I.; Barran, G.; Paterson, M.; Hilkens, C.M.U.; Lord, P. Tolerogenic dendritic cell reporting: Has a minimum information model made a difference? PeerJ. 2023, 11(e15352), 1-16.
101. Robertson, H.; Li, J.; Kim, H.J.; Rhodes, J.W. Harman, A.N.; Patrick, E.; et al. Transcriptomic Analysis Identifies A Toler-ogenic Dendritic Cell Signature. Front Immunol. 2021, 12(733231), 1-14.
103. Kim, H.J.; Park, J.S. Usage of Human Mesenchymal Stem Cells in Cell-based Therapy: Advantages and Disadvantages. Dev Reprod. 2017, 21(1), 1-10.
104. Thäte, C.; Woischwill, C.; Brandenburg, G.; Müller, M.; Böhm, S.; Baumgart, J. Non-clinical assessment of safety, biodis-tribution and tumorigenicity of human mesenchymal stromal cells. Toxicol Rep. 2021, 8, 1960-1969.
The changes are in section 5, table 1; section 5.1, lines 294-306; section 5.3, lines 333-336; and section 5.4, lines 354-358, and 368-371.
Comments 5: AI usage – Clarify validation of AI-generated figures for scientific accuracy and originality. Ensure compliance with journal standards.
Response 5: We appreciate the reviewer’s comment regarding the use of AI. As specified in Section 2, lines 111-113, AI (ChatGPT, GPT-4-turbo) was employed exclusively to refine the language and clarity of the text, and to generate the illustrative image of the inflamed joint included in Figures 1 and 2. The scientific content, accuracy, and interpretation of the figures were validated by the authors, who take full responsibility for their originality and correctness. We confirm that the use of AI fully complies with the journal’s standards and ethical guidelines.
Comments 6: Minor Revision
The introduction is long and occasionally repetitive; consider condensing background on conventional DMARDs and JAK inhibitor safety to keep focus on novel immunomodulatory strategies.
Response 6: We thank the reviewer for this helpful suggestion. In response, we have condensed the section of the Introduction describing conventional DMARDs and JAK inhibitor safety. The revised text now provides only the essential context regarding methotrexate as the anchor drug, the role of biologics and JAK inhibitors in patients with inadequate response, and recent regulatory restrictions following the ORAL Surveillance trial. This streamlining avoids unnecessary repetition and ensures that the focus of the Introduction remains on emerging immunomodulatory strategies.
The changes are in section 1, lines 60-69.
Comments 7: Figures 1 and 2: clarify whether they are original or adapted from prior publications. If AI-generated, ensure accuracy of molecular pathways.
Response 7: We thank the reviewer for this observation. Figures 1 and 2 are entirely original and were generated by the authors with the assistance of AI (ChatGPT, GPT-4-turbo), as detailed in Section 2 (Materials and Methods). All depicted interactions and molecular relationships were carefully cross-checked against the cited literature to ensure scientific accuracy. The authors confirm full responsibility for the originality and correctness of the figures.
The changes are in lines 131, and 222.
Comments 8: Typographical consistency: abbreviations (e.g., “Treg” vs “Tregs”) should be harmonised throughout.
Response 8: We thank the reviewer for this comment. The differences in abbreviations (e.g., “Treg” vs “Tregs”) were initially intended to reflect singular or plural forms, as is the case for other abbreviations such as MSC, TolDC, or EV. However, to enhance consistency and clarity throughout the manuscript, we have decided to replace “Tregs” with “Treg cells” in all instances.
The changes are in section 1.1, line 84; figure 1, lines 123, and 130-131; section 3.1, lines 133-134; section 3.3; lines 187, 197, and 199; section 4.1, line 236; section 5, line 283; table 1, line 287; section 5.2; lines 308, 310, 311-312, and 314; section 6.1, line 401, and in the list of abbreviations.
We sincerely thank the reviewer for their constructive suggestions, which have greatly improved the clarity, balance, and overall quality of our manuscript.
Reviewer 2 Report
Comments and Suggestions for Authors
I would like to thank the authors for carefully addressing the comments raised in the previous round. Their responses are clear and appropriate, and they satisfactorily resolve my earlier concerns.
Author Response
We sincerely thank the reviewer for their constructive suggestions, which have greatly improved the clarity, balance, and overall quality of our manuscript.